# Assessment of River-Sea Interaction in the Danube Nearshore Area (Ukraine) by Bioindicators and Statistical Mapping

**Anastasiia Snigirova** [1], **Yuliya Bogatova** [1] **and Sophia Barinova** [2,*]

1   Institute of Marine Biology of the NAS of Ukraine, 65048 Odessa, Ukraine; a.a.snigirova@onu.edu.ua (A.S.); Bogatova.Yu.I@nas.gov.ua (Y.B.)
2   Institute of Evolution, University of Haifa, Haifa 3498838, Israel
*   Correspondence: sophia@evo.haifa.ac.il

**Abstract:** There is a lack of understanding of the main drivers that form the picture of biological communities of transitional waters in deltaic ecosystems under the influence of terrestrial sources. Analysis of hydrochemical parameters in relation to phytoplankton communities in the Ukrainian part of the Danube coastal zone (in August 2018) is the focus of current work. The goal was to identify patterns in the distribution of environmental parameters (salinity and nutrients) in the area of the shipping channel through the Bystry arm, as well as to assess the state of water quality. The ecological bioindicators approach using modern statistical methods, and ecological mapping shows sufficient achievements in interpreting the results. The indicators of salinity (mesohalobes) had better describe the character of the transportation of fresh riverine waters than salinity gradient. The composition of 35 indicator phytoplankton species corresponds to 3 and 4 water quality classes in the coastal zone. High N:P ratios showed an imbalance in the ecosystem as an indicator of production and destruction processes. Statistical maps of the indicator species distribution revealed the river current's influence on the nearshore water mass. Ecological maps of surface and bottom variables show various environmental impacts resulting from dredging in the shipping channel and excavated soil dumping. Canonical correspondence analysis (CCA) and statistical maps revealed two pools of factors with oppositely directed effects on phytoplankton: salinity, on one hand, and nutrients, on the other. Miozoa and Chlorophyta have an opposite interaction with salinity and oxygen and can be ecosystem change indicators in further analysis

**Keywords:** Danube Delta; ecological mapping; nearshore area; nutrients; phytoplankton; transformation zones

## 1. Introduction

The Danube River plays a great role for the Black Sea by forming the hydrochemical regime in the northwestern shelf, which is the biggest shallow area in the sea with the high stocks of biological production. The Danube, together with the Dnieper and Dniester rivers, make up to 70% of the input in the northwestern part of the Black Sea [1–3]. Understanding the processes of interaction of one of the largest rivers of Europe with the sea is of great importance.

The Danube nearshore area represents a transitional river-sea contact zone. It is formed due to dynamic interaction, mixing, and transformation of the river's water masses and the sea. The Ukrainian part of the nearshore area covers the region from the edge of the Kiliya Arm (Kiliya Delta of the Danube River) to the seawaters' border with a salinity of 17‰. The nearshore outer border is conditional because its location depends on the runoff rate of the Danube River and the wind regime [1,2]. The vertical distribution of waters at the nearshore area is characterized by sharp salinity changes and temperature, especially in spring and early autumn. It is noted a wide variability and fractal in the distribution of hydrochemical and biological parameters. They are associated with the seasonal variability of river runoff in the nearshore, which is the primary source of nutrients

that ensure the seasonal development of production processes [2]: the special hydrological-hydrochemical and hydrobiological regimes in the transition waters of Danube nearshore results in increased biological productivity. The Danube water flowing to the seaside occupies its upper layer up to 5 m. As a rule, these are seaside areas located directly near the mouths of large delta's arms. With a distance from the mouth, the river water gradually transforms into seawater, passing through three successive zones: the principal (salinity up to 10‰), frontal (10–12‰), and final (12–17‰) [3].

There is only one navigable arm in the Ukrainian part of the Kiliya Danube Delta—the Bystryi Arm. To maintain the Danube–Black Sea channel's working depths through the Bystryi Arm, periodically dredging operations are carried out. These operations' impact is overlapped with the river-sea interactions that form the main background in these transition waters. The salinity of the nearshore water masses at a depth of more than 5 m makes 16–18‰. The dumping area is located at a distance of 8 km from the delta edge, in the final zone. Since 2004, the Institute of marine biology of the NAS of Ukraine organizes environmental monitoring to determine the impact of these manipulations on the ecotone marine-river ecosystem [4,5]. However, there is still a lack of a general description of the ecosystem functioning and understanding of how river-sea interactions influence biological communities.

The studies of 2004–2011 describe phytoplankton's biodiversity in the Ukrainian part of the Danube nearshore area with 409 taxa [6,7]. The most remarkable species richness is typical for diatoms (155 species) and green (122) algae. They are followed by dinophytes (70) and cyanobacteria (34). A small number of taxa represents golden, euglenoid, and cryptophyte algae. Compared to the Romanian part of the Danube Delta, 957 species of microalgae are indicated, excluding the section of green ones, which include another 526 species [8].

The monitoring of water bodies, especially assessing the anthropogenic pressure, implies following the European directives' requirements (Water Framework Directive and Marine Strategy Framework Directive) [9,10], using different approaches to assessing the quality of the aquatic environment [11,12]. One of the most effective methods for water quality assessment is bioindication based on species composition shifts as a response of biota to the environmental changes and influence from land base sources [13–15].

In the Black Sea and many other marine water areas, species indicators' elaboration suitable for environmental monitoring is in the initial stage [16–18]. The level of nutrients and chlorophyll concentrations is widely used as an indicator of the eutrophication assessment [19]. In the northwestern part of the Black Sea and western part of the Azov Sea, the bioindication method was started for benthic species [20–23]. Colleagues from Ukraine, working within the framework of the development of a national marine strategy, are actively introducing a morphological and functional approach to assess the state of the environment using indices of the specific surface area of macrophytes and microalgae [24–26]. Integrated indices and comparative statistics, using various structural indicators of marine and estuarine ecosystems, is widespread [27,28].

Modern statistical programs and mapping have already been tested for many freshwater objects in different climatic zones [13–15,29–31]. However, applying these approaches to assessing the state of marine ecosystems has not yet received widespread use due to the limited number of indicator species [20,21,32]. The combined approach of studying the species composition of algae and statistics in the coastal area has good examples of its application [28]. We believe it rational to apply this complex of methods for the Danube nearshore area—a dynamically changing ecosystem, using phytoplankton's structural indicators and the content of nutrients. In addition, we think it is essential to contribute to the development of the database on marine species indicators that could be the base for assessing the ecological class of the water areas.

We hypothesize that it is possible to reveal the Danube Delta waters' influence on the coastal ecosystems of the Black Sea using bioindication of phytoplankton communities and statistical methods.

The work aimed to assess the hydrochemical state and phytoplankton community of the Ukrainian part of the Danube nearshore area in the summer employing statistical analysis and ecological mapping. This work will be the base for the long-term analysis to identify the patterns in the distribution of environmental parameters and phytoplankton in the studied region.

## 2. Materials and Methods

The materials used in the study were obtained during environmental monitoring at the Ukrainian part of the nearshore area of the Danube Delta in August 2018 (Figure 1). Water samples for the determination of hydrochemical parameters and phytoplankton were simultaneously taken in the surface and bottom layers using a Niskin bathometer. Determination of hydrochemical parameters: salinity, dissolved oxygen and% of its saturation, and pH value were carried out onboard the vessel. Dissolved inorganic nitrogen (DIN = $NH_4^+$ + $NO_2^-$ + $NO_3^-$), dissolved organic nitrogen (DON), total nitrogen (NT), dissolved inorganic phosphorus (DIP = $PO_4^{3-}$), dissolved organic phosphorus (DOP), total phosphorus (PT), and silicic acid ($SiO_3^{2-}$), total suspended matter (TSS) in water samples were determined in a stationary laboratory of the Institute of Marine Biology of the NAS of Ukraine by standard methods accepted in international practice [33].

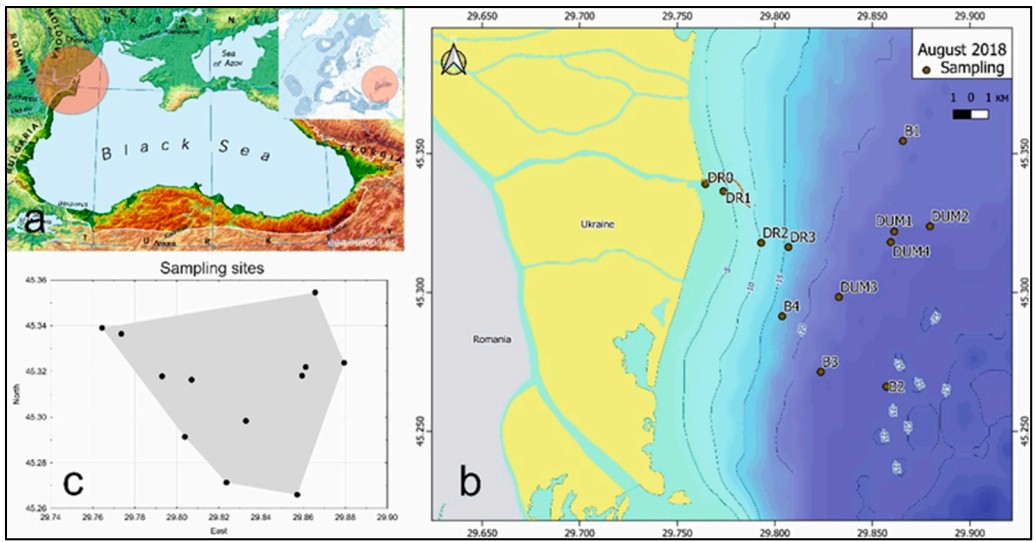

**Figure 1.** Sampling sites in the Ukrainian part of Danube nearshore area in August 2018 (**a**,**b**). Station names (**b**): DR—drainage area, DUM—dumping area, B—background area. Mapped area (gray) in the Statistica 12.0 program with sampling sites as black dots (**c**).

Phytoplankton samples were fixed in 4% neutral formaldehyde and concentrated by sedimentation method to a volume of 40–80 mL. Sample processing was carried out using a Konus Biorex microscope at a magnification of $\times$160–640. Cells were counted in a 0.05 mL counting chamber. To calculate the biomass, we used the "true" cell volumes calculated on the basis of measurements of the linear dimensions of the cells [34]. For species identification and nomenclature, we used the following sources [35–39].

The index of saprobity was chosen as it reflects the organic pollution and includes not only diatoms but other algae that are rather represented in studied river-mouth communities [40,41]. Indices saprobity (S) were calculated based on identified species for each community and quantitative investigations of phytoplankton as:

$$S = \sum_{i=1}^{n}(s_i \times a_i) / \sum_{i=1}^{n}(a_i) \tag{1}$$

where *S*—saprobity index of the algal community; $s_i$—species-specific saprobity index; $a_i$—species abundance.

The ecosystem state index WESI was calculated for each community based on the classification ranks of the nitrate-nitrogen content and the index of saprobity of the community [13]. The classes of water quality were given according to [31] (Table 1). The sites that were included in the analysis are distinguished according to the tasks of ecological monitoring: dredging operations on the offshore approach channel to the mouth of the Bystryi Arm with depths of 0–10 m (sites DR0, 1, 2, 3); dumping area of dredging soil with depths of 23–25 m (sites DUM 1, 2, 3, 4); background areas to the north and south of the mouth of the Bystryi Arm and dumping area (sites B1, 2, 3, 4) with depths of 17–23 m.

**Table 1.** Determination of water quality (EU color codes) based on saprobity index (S) according to [31].

| Class of Water Quality | Saprobity Index (S) | Water Quality |
|---|---|---|
| I | 0–0.5 | Very good |
| II | 0.5–1.5 | Good |
| III | 1.5–2.5 | Fair |
| IV | 2.5–3.5 | Fairly poor |
| V | 3.5–4.0 | Poor |
| VI | >4.0 | Very poor |

Environmental mapping was carried out in the Statistica 12.0 program according to the parameter values and geographic coordinates of each site. Canonical correspondence analysis (CCA) was done with the CANOCO Program 4.0 [42], calculation of similarity was doing as the network analysis in JASP on the botnet package in R Statistica package of [43].

## 3. Results

### 3.1. Hydrological and Hydrochemical Parameters

The distribution of hydrochemical parameters at the Danube nearshore area in August 2018 was characterized by significant spatial variability, which is associated with both the two-layer structure of the waters and the development of production-destruction processes (Tables 2 and A1; Figures 2–6).

**Table 2.** Average data of the depth, salinity and nutrients in the Ukrainian part of Danube nearshore area in August 2018.

| Region | Horizon | Depth, m | S, ‰ | DIN | NT | DIP | PT | NT:PT |
|---|---|---|---|---|---|---|---|---|
| | | | | $\mu$gN L$^{-1}$ | | $\mu$gP L$^{-1}$ | | |
| Drainage area (DR) | Surface | 0 | 2.7 | 910 | 3000 | 50.1 | 64.8 | 46:1 |
| | Bottom | 6.4 | 9.3 | 960 | 3600 | 44.1 | 50.0 | 72:1 |
| Dumping area (DUM) | Surface | 0 | 15 | 63 | 760 | 6.3 | 22.4 | 34:1 |
| | Bottom | 20 | 17.9 | 32 | 550 | 6.7 | 20.4 | 27:1 |
| Background area (B) | Surface | 0 | 12.5 | 170 | 1400 | 11.4 | 30.0 | 47:1 |
| | Bottom | 20.2 | 17.8 | 42 | 1900 | 12.0 | 27.0 | 70:1 |
| Average | Surface | - | - | 380 | 1720 | 22.6 | 38.8 | 44:1 |
| | Bottom | - | - | 345 | 2017 | 21.0 | 32.5 | 62:1 |

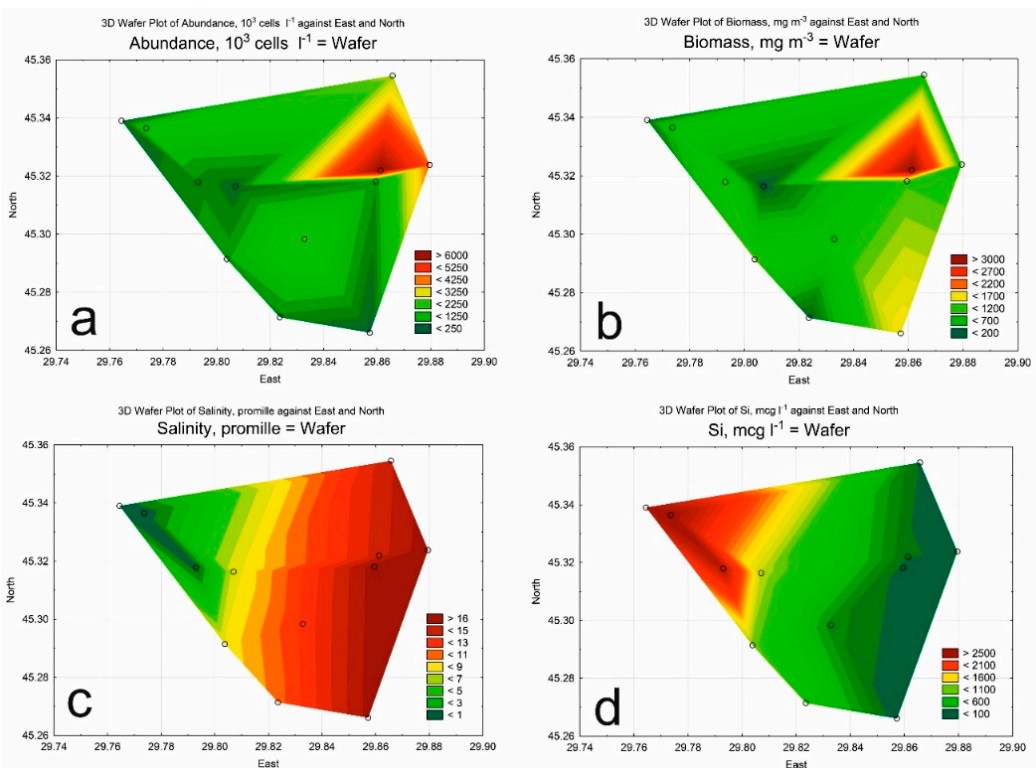

**Figure 2.** Statistically generated maps of phytoplankton (abundance (**a**), biomass (**b**)) and hydrological and chemical parameters of water (salinity (**c**), silicium (**d**)) on the surface horizon in the Ukrainian part of the Danube nearshore area in August 2018.

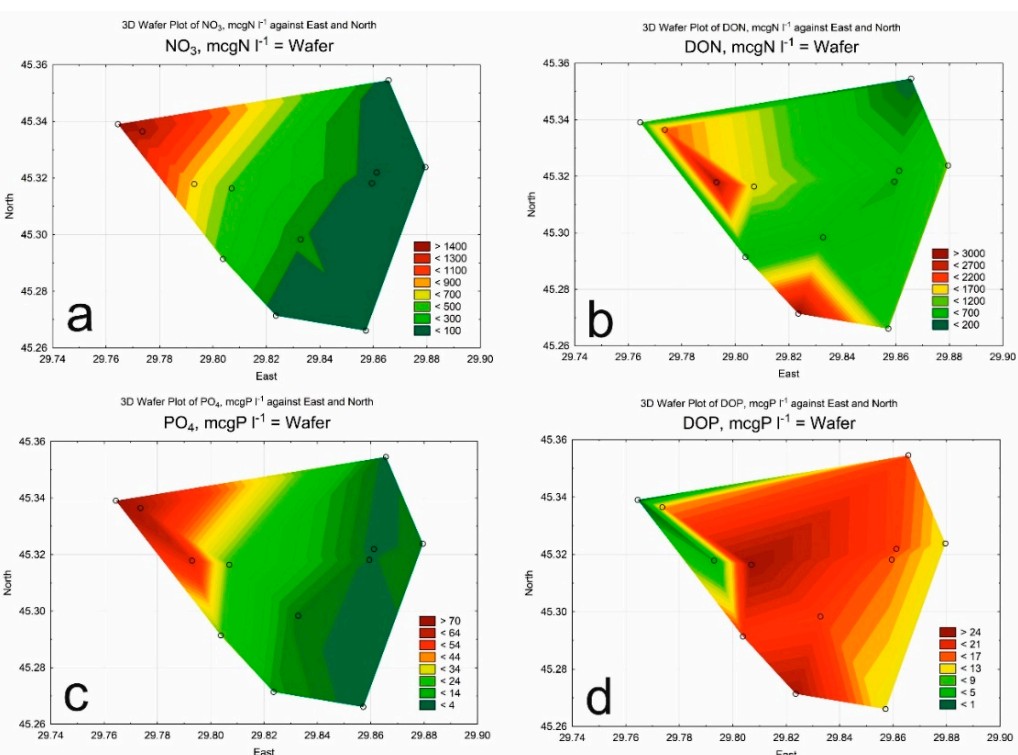

**Figure 3.** Statistically generated maps of nutrients from the surface horizon in the Ukrainian part of the Danube nearshore area in August 2018: nitrates ($NO_3^-$) (**a**); dissolved organic nitrogene (DON) (**b**); dissolved inorganic phosphorus (DIP = $PO_4^{3-}$) (**c**); dissolved organic phosphorus (DOP) (**d**).

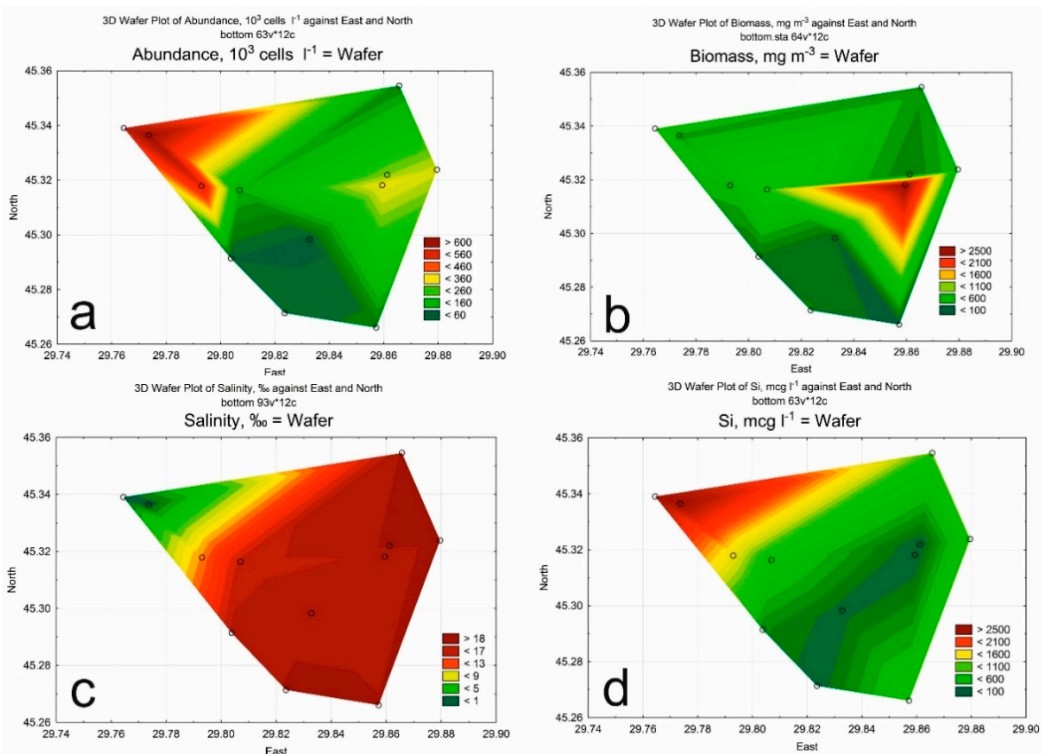

**Figure 4.** Statistically generated maps of phytoplankton abundance (**a**) and biomass (**b**) and hydrological and chemical parameters of water (salinity (**c**), silicium (**d**)) on the bottom horizon in the Ukrainian part of the Danube nearshore area in August 2018.

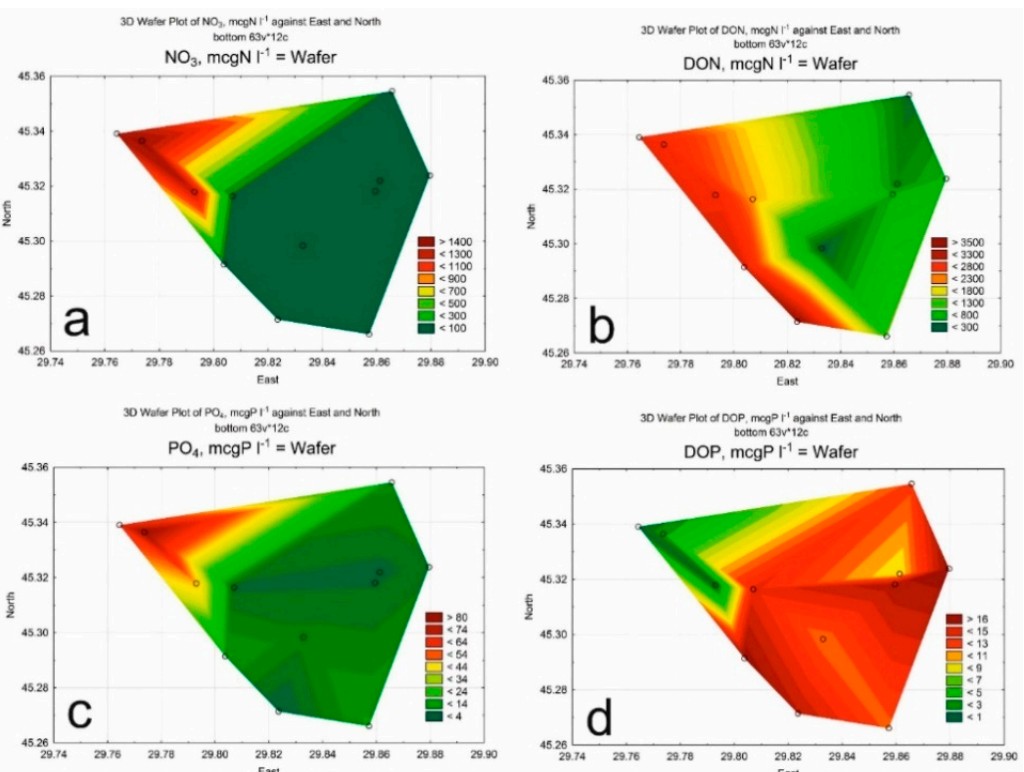

**Figure 5.** Statistically generated maps of nutrients from the bottom horizon in the Ukrainian part of the Danube nearshore area in August 2018: nitrates ($NO_3^-$) (**a**); dissolved organic nitrogene (DON) (**b**); dissolved inorganic phosphorus (DIP = $PO_4^{3-}$) (**c**); dissolved organic phosphorus (DOP) (**d**).

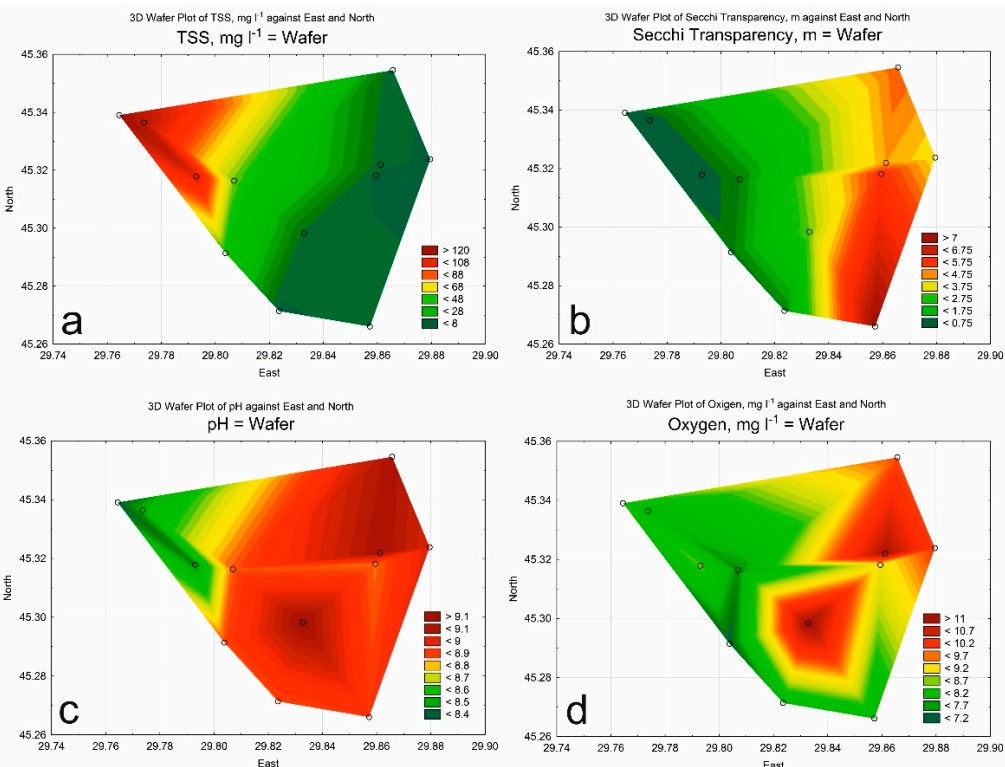

**Figure 6.** Statistically generated maps of suspended substance (TSS) (**a**), water transparency (Secchi) (**b**), pH (**c**) and oxygen compound (**d**) from the surface horizon in the Ukrainian part of the Danube nearshore area in August 2018.

The maps show the distribution of salinity, silicic acid (Figures 2c and 4c) and different forms of nutrients (Figures 3 and 5). The content of nutrients at the nearshore area was quite high: $NH^{4+}$ up to 53.5 $\mu gN\ L^{-1}$, DIP up to 80.4 $\mu gP\ L^{-1}$. In the surface layer of the nearshore area, with a distance from the delta edge and increasing salinity, a significant decrease in the content of $SiO_3^{2-}$, $NO_3^{-}$, $PO_4^{-}$ was noted, which confirms the input of mineral compounds with river runoff (Figure 3a,c and Figure 5a,c). An increase of the DOP content in the surface layer with an increase in salinity indicates the active development of production and destruction processes at the nearshore (Figure 3d). In the bottom layer, formed by water masses of marine genesis, with the increasing of depths, there was no significant accumulation of DOP and DON (Figure 5b,d).

The distribution of total suspended substance (TSS) and gradient of water transparency (Secchi) is demonstrated in Figure 6a,b showing the transition of particles in the surface layers from the river to sea. The range of variability of dissolved oxygen in the surface layer was 93–139% of saturation, pH—8.39–9.12 (Figure 6c,d). In the near-bottom horizon at station B2, the water was not enough saturated with oxygen (41%).

The ratio NT: PT varied from 34:1 to 47:1 and from 27:1 to 72:1 in the surface and bottom layers, respectively, in different parts of the nearshore area in August 2018 (Table 2, Figure 7e,f).

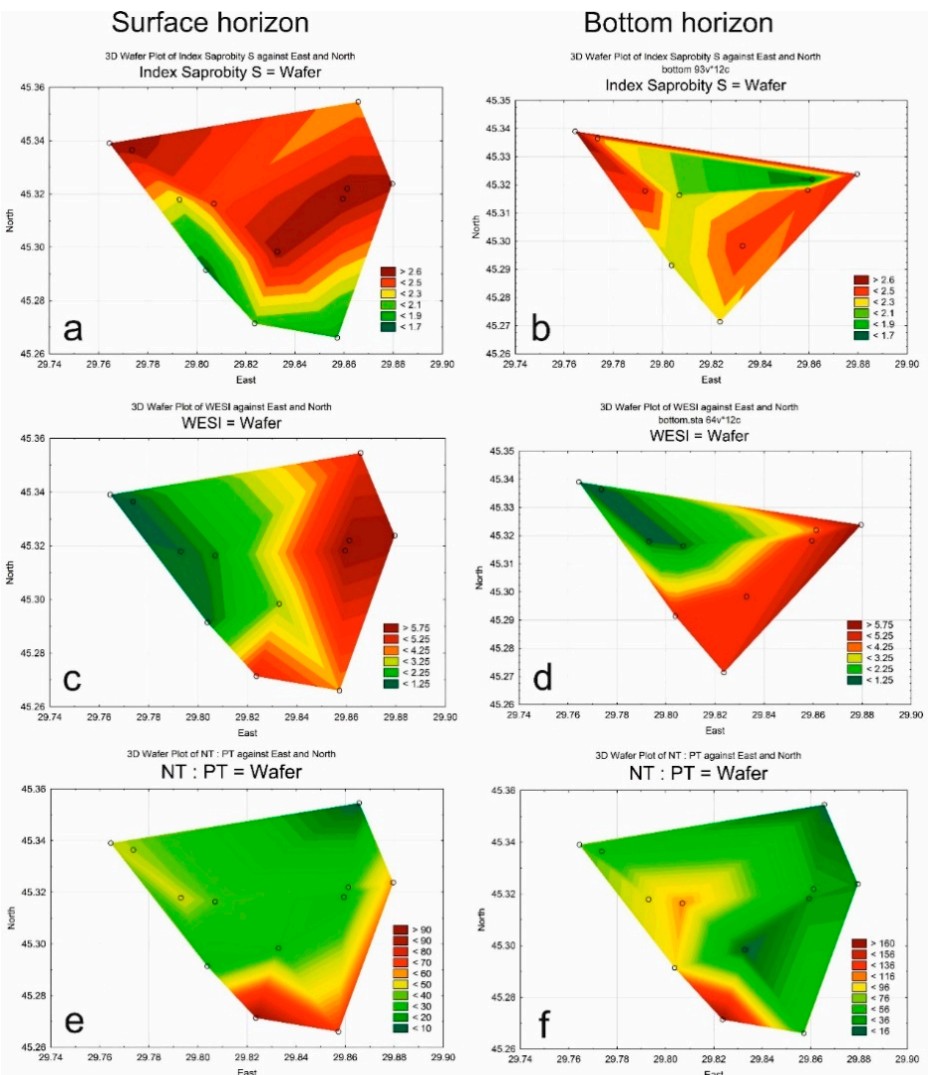

**Figure 7.** Statistically generated maps of the index of saprobity (S) (**a**,**b**), index of ecosystem state (WESI) (**c**,**d**) based on phytoplankton species and relation N:P (**e**,**f**) in the Ukrainian part of Danube nearshore area in August 2018: (**a**,**c**,**e**)—surface horizon; (**b**,**d**,**f**)—bottom horizon.

*3.2. Phytoplankton*

In August 2018, in the phytoplankton assemblage, the representatives of six divisions of microalgae were registered: Bacillariophyta, Miozoa, Chlorophyta, Cyanobacteria, Cryptophyta, and Euglenozoa. The most widespread groups were Bacillariophyta and Miozoa, forming an average of 66.9% and 28.8% of total biomass, respectively. A total of 66 species were identified: Bacillariophyta—32, Miozoa—16, Chlorophyta—7, Cyanobacteria—3, Cryptophyta—3, Euglenozoa—2, small flagellates—3 species.

Between the diatoms, *Cyclotella caspia* Grunow 1878, *C. meneghiniana* Kützing, *Pseudo-nitzschia delicatissima* (Cleve) Heiden, *Pseudosolenia calcar-avis* (Schultze) B.G.Sundström were met frequently. Between the Miozoa *Prorocentrum cordatum* (Ostenfeld, 1901) Dodge, *Prorocentrum micans* Ehrenberg. We also noted the diatoms *Proboscia alata* (Brightwell) Sundström, *Skeletonema costatum* (Greville) Cleve, *S. subsalsum* (Cleve-Euler) Bethge, *Cerataulina pelagica* (Cleve) Hendey; dinoflagellates *Protoperidinium divergens* (Ehrenberg, 1841) Balech, *Tripos furca* (Ehrenberg) F.Gómez, *T. fusus* (Ehrenberg) F.Gómez, *Scrippsiella trochoidea* (Stein, 1883) Balech ex Loeblich. Chlorophyta was met not very often: *Monoraphidium arcuatum* (Korsch.) Hindak, *M. minutum* (Nägeli) Komárková-legnerová, *M. griffithii* (Berkeley) Komárková-Legnerová, *Actinastrum hantzschii* Lagerh. Cyanobacteria were registered very

rarely—just on three sites, in the dredging and dumping area and were presented with two species (Table 3).

**Table 3.** List of species in the phytoplankton in the Ukrainian part of Danube nearshore area in August 2018 with some ecological preferences.

| Taxa | Surf | Bott | Hab | Halob | Sap | SLA |
|---|---|---|---|---|---|---|
| **Bacillariophyta** | | | | | | |
| *Actinocyclus normanii* (W.Gregory ex Greville) Hustedt 1957 | 1 | 1 | P | mh | b | 2.3 |
| *Aulacoseira islandica* (O.Müller) Simonsen, 1979 | 1 | 0 | P-B | i | b-o | 1.6 |
| *Cyclotella caspia* Grunow 1878 | 1 | 1 | P | eu | - | - |
| *Cyclotella meneghiniana* Kützing, 1844 | 1 | 1 | P-B | hl | a-o | 2.8 |
| *Lauderia confervacea* Cleve, 1896 | 1 | 0 | P | eu | - | - |
| *Diatoma tenuis* C.A. Agardh, 1812 | 1 | 1 | P-B | hl | o | 1.3 |
| *Leptocylindrus danicus* P.T. Cleve, 1889 | 0 | 1 | P | eu | - | - |
| *Melosira moniliformis* (O.F. Müller) C. Agardh, 1824 | 1 | 0 | P-B | hl | b | 2 |
| *Navicula cryptocephala* Kützing, 1844 | 0 | 1 | P-B | i | b | 2.1 |
| *Pantocsekiella kuetzingiana* (Thwaites) K.T.Kiss and E.Ács | 1 | 1 | P-B | i | b | 2.1 |
| *Proboscia alata* (Brightwell) Sundström, 1986 | 1 | 1 | P | eu | - | - |
| *Pseudo-nitzschia pseudodelicatissima* (G.R. Hasle) G.R. Hasle 1993 | 1 | 1 | P | eu | - | - |
| *Pseudosolenia calcar-avis* (Schultze) B.G.Sundström, 1986 | 1 | 1 | P | eu | - | - |
| *Skeletonema costatum* (Greville) P.T. Cleve, 1878 | 1 | 1 | P | eu | - | - |
| *Skeletonema subsalsum* (A.Cleve) Bethge, 1928 | 0 | 1 | P | eu | - | - |
| *Tabularia fasciculata* (C.Agardh) D.M.Williams and Round, 1986 | 1 | 1 | P-B | mh | b-a | 2.5 |
| **Miozoa** | | | | | | |
| *Prorocentrum balticum* (Lohmann, 1908) Loeblich, 1970 | 1 | 0 | P | eu | - | - |
| *Prorocentrum cordatum* (Ostenfeld, 1901) Dodge, 1975 | 1 | 1 | P | eu | - | - |
| *Prorocentrum micans* Ehrenberg, 1834 | 1 | 1 | P | eu | - | - |
| *Prorocentrum lima* (Ehrenberg, 1860) Stein, 1975 | 1 | 1 | P | eu | - | - |
| *Protoperidinium divergens* (Ehrenberg, 1841) Balech, 1974 | 1 | 1 | P | eu | - | - |
| *Scrippsiella aqcuminata* (Ehrenberg) Kretschmann, Elbrächter, Zinssmeister, S.Soehner, Kirsch, Kusber and Gottschling 2015 | 1 | 0 | P | eu | - | - |
| *Tripos furca* (Ehrenberg) F.Gómez, 2013 | 1 | 1 | P | eu | - | - |
| *Tripos fusus* (Ehrenberg) F.Gómez, 2013 | 0 | 1 | P | eu | - | - |
| **Chlorophyta** | | | | | | |
| *Actinastrum hantzschii* Lagerheim, 1882 | 0 | 1 | P-B | i | b | 2.3 |
| *Kirchneriella lunaris* (Kirchner) K. Möbius, 1894 | 1 | 0 | P-B | i | o-a | 1.8 |
| *Monactinus simplex* (Meyen) Corda 1839 | 1 | 0 | P-B | - | b | 2 |
| *Monoraphidium arcuatum* (Korshikov) Hindák, 1970 | 1 | 1 | P-B | i | b | 2.1 |
| *Monoraphidium contortum* (Thuret) Komárková-Legnerová 1969 | 0 | 1 | P-B | i | b | 2.2 |
| *Monoraphidium griffithii* (Berkeley) Komárková-Legnerová, 1969 | 0 | 1 | P-B | i | b | 2.2 |
| *Monoraphidium minutum* (Nägeli) Komárková-legnerová 1969 | 1 | 0 | P-B | i | b-a | 2.5 |
| **Cryptophyta** | | | | | | |
| *Cryptomonas erosa* Ehrenberg, 1832 | 0 | 1 | P | - | b | 2.3 |
| *Hillea fusiformis* (J.Schiller) J.Schiller 1925 | 0 | 1 | P | eu | - | - |
| **Cyanobacteria** | | | | | | |
| *Jaaginema kisselevii* (Anissimova) Anagnostidis and Komárek, 1988 | 1 | 0 | P-B | mh | - | - |
| *Phormidium nigroviride* (Thwaites ex Gomont) Anagnostidis, Komárek, 1988 | 0 | 1 | P-B | eu | - | - |

Note: surf—surface horizon, bott—bottom horizon; Hab—indicators of habitat preference: P—planktic. P.-B.—plankto-benthic species; Halob—salinity preference: i—oligohalobes-indifferents, mh—mesohalobes, hL—halophiles, eu—marine, euhalobes; Sap—indicators of self-purification zone: o—oligosaprobes; o-a—oligo-alpha-mesosaprobes, b-o—beta-oligosaprobes, b-a—beta–alpha-mesosaprobes, b—beta-mesosaprobes, a-o—alpha-oligosaprobes; SLA—species-specific index of saprobity according to Sládeček.

Out of the 66 species found, 35 were indicative (Table 3), with almost half of the species having an indicative role only in relation to salinity and habitat. Our bioindication results show that most species are planktonic (54.3%) and plankto–benthic (45.7%). More than half of the species (54.5%) were marine, and 27.3% oligohalobes–indifferents.

The highest values of the abundance and biomass of phytoplankton in surface layers were registered in the zone of dumping with maximum on-site DUM1 (Table 4, Figure 2a,b). Biomass and abundance of phytoplankton in other regions were distributed homogeneously.

**Table 4.** Structural parameters of phytoplankton on two horizons in the Ukrainian part of the Danube nearshore area in August 2018.

| Site | Surface | | | | Bottom | | | |
|---|---|---|---|---|---|---|---|---|
| | Abundance, $10^3$ cells $L^{-1}$ | Biomass, mg m$^{-3}$ | Index Saprobity SLA | Index WESI | Abundance, $10^3$ cells $L^{-1}$ | Biomass, mg m$^{-3}$ | Index Saprobity SLA | Index WESI |
| B1 | 2591.62 | 966.31 | 2.32 | 5.00 | 101.35 | 333.93 | - * | - |
| B2 | 150.53 | 1700.56 | 1.80 | 4.00 | 86.96 | 45.02 | - | - |
| B3 | 672.04 | 162.60 | 2.03 | 5.00 | 69.84 | 209.00 | 2.25 | 5.00 |
| B4 | 936.28 | 1147.21 | 1.69 | 1.33 | 59.47 | 175.00 | 2.10 | 5.00 |
| Average | 1087.62 | 994.17 | 1.96 | 3.83 | 79.41 | 190.74 | 2.18 | 5.00 |
| DR0 | 263.77 | 279.02 | 2.80 | 1.20 | 597.32 | 656.07 | 2.80 | 1.20 |
| DR1 | 1724.56 | 1015.96 | 2.75 | 1.20 | 663.99 | 365.27 | 2.32 | 1.00 |
| DR2 | 702.39 | 1161.06 | 2.20 | 1.25 | 554.46 | 497.07 | 2.53 | 1.20 |
| DR3 | 324.26 | 125.13 | 2.45 | 1.67 | 141.79 | 899.67 | 2.10 | 1.25 |
| Average | 753.75 | 645.29 | 2.55 | 1.33 | 489.39 | 604.52 | 2.44 | 1.16 |
| DUM1 | 6683.35 | 3342.33 | 2.78 | 6.00 | 278.97 | 195.12 | 1.66 | 4.00 |
| DUM2 | 4302.36 | 1031.39 | 2.75 | 6.00 | 310.86 | 508.79 | 2.61 | 6.00 |
| DUM3 | 2288.47 | 995.41 | 2.76 | 3.00 | 53.44 | 101.01 | 2.50 | 5.00 |
| DUM4 | 793.99 | 1177.15 | 2.80 | 6.00 | 332.71 | 2764.94 | 2.43 | 5.00 |
| Average | 3517.04 | 1636.57 | 2.77 | 5.25 | 244.00 | 892.47 | 2.30 | 5.00 |

*—the absence of specific indicators of saprobity.

In the bottom layers, the situation is different, showing the maximum growth of phytoplankton abundance in the river mouth area (sites DR0, DR1, DR2), whereas the maximum biomass was noted in site DUM4. This was due to the presence of dinoflagellates (Miozoa) which results in high biomass because of big cell volume.

The abundance of phytoplankton in the surface layers of the dredging area was twice higher than on the bottom one, whereas, in other areas, these discrepancies were much more considerable (Table 4). The diatoms dominated (523.1 mg m$^{-3}$), their contribution to the total phytoplankton biomass made 63.2%. Biomass of Miozoa made 303.2 mg m$^{-3}$ or 33.6%, Chlorophyta—1.7 mg m$^{-3}$ (0.2%), Cyanobacteria—8.2 mg m$^{-3}$ (1.0%). Other groups of microalgae met rarely; their biomass did not exceed 1.5%.

In Figure 8a, the indicators of salinity—mesohalobes (mh) that survive in brackish waters demonstrate the character of the distribution of riverine freshwaters in the nearshore area. The presence of mesohaline (mh) species demonstrates dividing the nearshore area into three water masses to the south from the mouth, the north one, and the south one. Species-indicators of class 3 of water quality (Figure 8b) and nitrogen-autotrophic taxa (ate) (Figure 8c) represent the spreading of riverine waters within the marine water. Maximums of plankto-benthic species (Figure 8d) are also associated with the character of water flow from the river. Waters from the dumping area with a bigger amount of plankto-benthic species are transported with intensive water flow to the background sites.

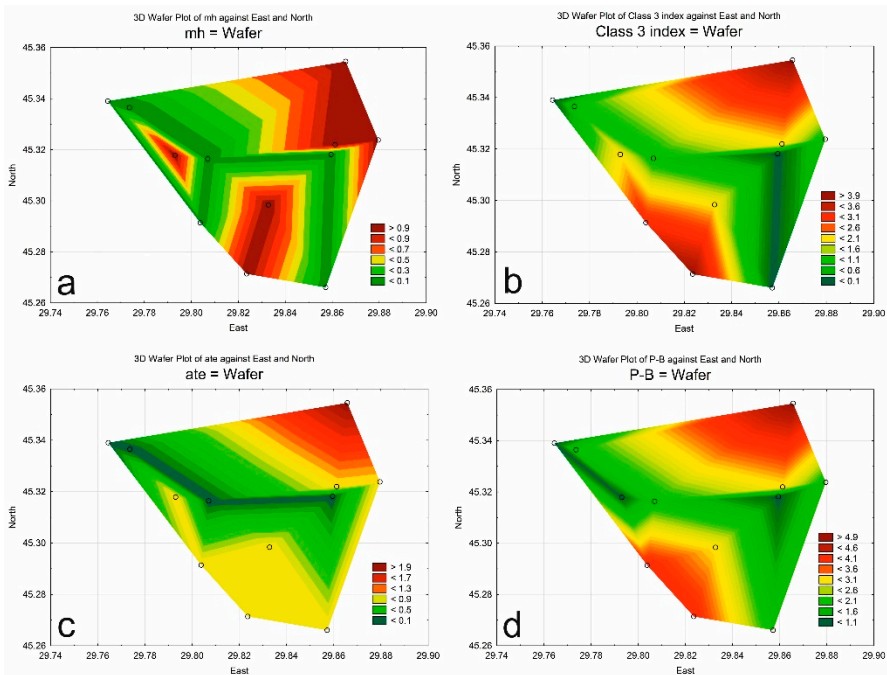

**Figure 8.** Statistically generated maps of mesohalobe species (mh) (**a**), species-indicators of class 3 of water quality defined by species-specific index SLA value (**b**), nitrogen-autotrophic taxa (ate), tolerating elevated concentrations of organically bound nitrogen (**c**), plankto–benthic species (P–B), (**d**) from the surface horizon in the Ukrainian part of Danube nearshore area in August 2018.

### 3.3. Indices

Analyzing the distribution of the index of saprobity, we see the input of the organic nitrogen and phosphorus from the area of dredging (Figure 7). Figures show two pools of available organic nitrogen: in the mouth of the Bystryi Arm and in the deeper area, both in the surface (Figure 7a) and bottom (Figure 7b) layers of water. The first maximum of DON is explained by its release from the bottom sediments during the dredging, whereas the second one is caused by the photosynthetic activity of microalgae in the surface area and destruction in the bottom area. The average values of the index of saprobity on the surface layer made 2.4 (1.69–2.80), and on the bottom—2.3 (1.66–2.80) (Table 4), which corresponds in average to the class 3 of water quality (beta-mesosaprobity), but in the mouth of the river it increases up to class 4. The data on the water quality class are given in Table 5.

**Table 5.** Number of species indicating the affiliation to water quality class by the index of saprobity (SLA).

| Sites | B1 | B2 | B3 | B4 | DR0 | DR1 | DR2 | DR3 | DUM1 | DUM2 | DUM3 | DUM4 | Total |
|-------|----|----|----|----|-----|-----|-----|-----|------|------|------|------|-------|
| Surface | | | | | | | | | | | | | |
| Class 2 | 1 | 1 | 0 | 0 | 0 | 0 | 0 | 0 | 0 | 0 | 0 | 0 | 2 |
| Class 3 | 4 | 0 | 4 | 3 | 0 | 1 | 2 | 1 | 0 | 1 | 2 | 0 | 18 |
| Class 4 | 0 | 1 | 0 | 1 | 1 | 1 | 0 | 1 | 0 | 1 | 1 | 1 | 8 |
| Bottom | | | | | | | | | | | | | |
| Class 2 | 0 | 0 | 0 | 0 | 0 | 0 | 0 | 0 | 1 | 0 | 0 | 0 | 1 |
| Class 3 | 0 | 0 | 2 | 1 | 0 | 4 | 3 | 1 | 1 | 1 | 1 | 1 | 15 |
| Class 4 | 0 | 0 | 0 | 0 | 1 | 1 | 1 | 0 | 1 | 1 | 0 | 1 | 6 |

The index of the negative impact on the aquatic ecosystem (WESI) turned out to be much higher than 1, which indicates the ability of the ecosystem of the Danube nearshore area to counteract factors negative for photosynthesis. However, in Figure 7c,d, it can be seen that the lowest index values are associated with dredging at both surface and bottom layers.

Moreover, the maximum of quantitative characteristics of the phytoplankton on the surface horizon coincides with the level of saprobity index and WESI index. The phytoplankton distribution on the bottom layer in its turn coincides with the saprobity index that represents the two spots of microalgae growth: in the river mouth area and in the zone where the soil from the dredging area was dumped.

### 3.4. Diagrams and Clusters

The relation of the quantitative development of phytoplankton with the main environmental parameters on the surface and in the bottom layer is shown in Figure 9a,b. For both horizons, we mention two vectors of the distribution patterns of the factors: the effect of salinity on phytoplankton was opposite to the nutrients. Index of saprobity (SLA) and WESI demonstrate the influence of environmental parameters (organic and toxic contamination) on the phytoplankton community, so they are included in the CCA.

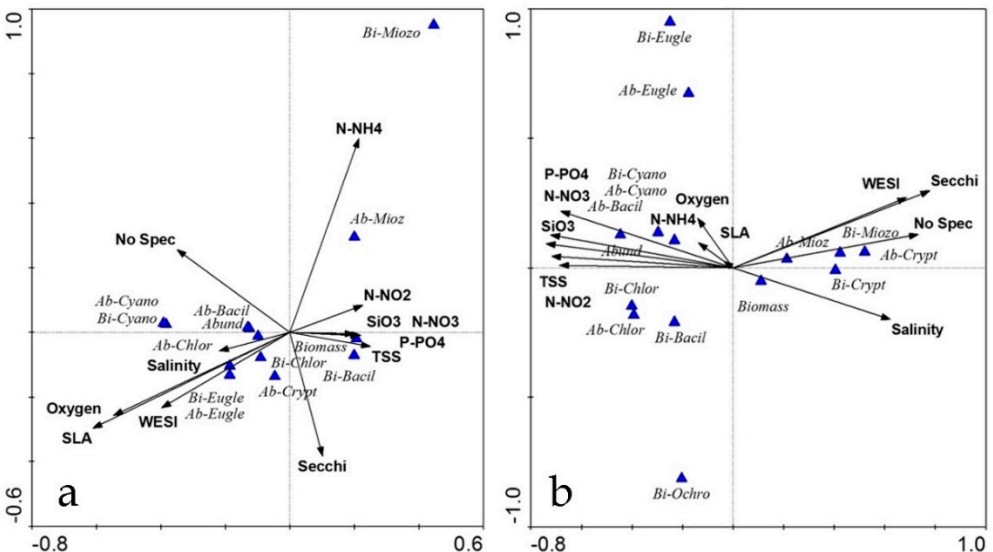

**Figure 9.** Ordination diagram based on the canonical correspondence analysis (CCA) analysis of species abundance and biomass on the surface (**a**) and bottom (**b**) horizon. Significance of the effect of six environmental variables: salinity, oxygen, nitrites (N-NO$_2$), nitrates (N-NO$_3$), ammonium (N-NH$_4$), orthophosphate acid (P-PO$_4$), silicic acid (SiO$_3$), total suspended substance (TSS), transparency (Secchi), number of species (No Spec), SLA (index of saprobity), WESI which is verified by Monte Carlo permutation test, $p < 0.0182$ (**a**), $p < 0.0146$ (**b**). The length and direction of the arrows indicate the relative significance and direction of change in environmental variables. Blue triangles—algae taxa (Bi—biomass, Ab—abundance).

CCA analysis showed that the most indicative groups of surface phytoplankton were dinoflagellates (Miozoa), which density increased at the surface in low oxygen concentrations and higher ammonium (N-NH$_4$). The opposite groups of phytoplankton were cyanobacteria and Euglenophyta that are managed by low nutrients and higher oxygen, salinity and SLA. On the surface layer, two more vectors were presented: ammonium (N-NH$_4$) and the number of species (No Spec) opposite to transparency (Secchi) (Figure 9a). The growth of diatoms in the surface horizon matches with the green algae and regulated by nutrients contrary to dinoflagellates.

Figure 9b demonstrates that in the bottom layer, the environmental parameters influencing the production of phytoplankton biomass were divided into two clusters, 1) salinity, transparency (Secchi), WESI and 2) nutrients and oxygen. Green and diatom algae, as well as cyanobacteria, react most actively to factor group 2. At the same time, dinoflagellates and Cryptophyta developed in conditions of increasing salinity, transparency (Secchi),

WESI and decreasing nutrients and oxygen. The suspended substance (TSS) parameter was united with nutrients both on the surface and bottom layer.

Figure 10 shows the similarity of studied parameters on monitoring sites ($p < 0.05$). First, we see that in surface layer background sites have similarities both to dredging and to dumping sites. This explains their different geographical location. The data are divided into 4 separate clusters (Figure 10a); two of them are closer to one another—these are dredging areas and two background sites (B3, B4). They can be united in one cluster because of light links between them and due to joint hydrodynamic and hydrological conditions under the influence of the river discharge (Figure 1). Clusters 3 and 4 in Figure 10a show the similarity of the dumping area and two other background sites (B1, B2) that are located in much more deep areas of the nearshore area of the Danube. The strong positive links in the surface horizon were mentioned for DUM4 and B2; DR1 and DR0.

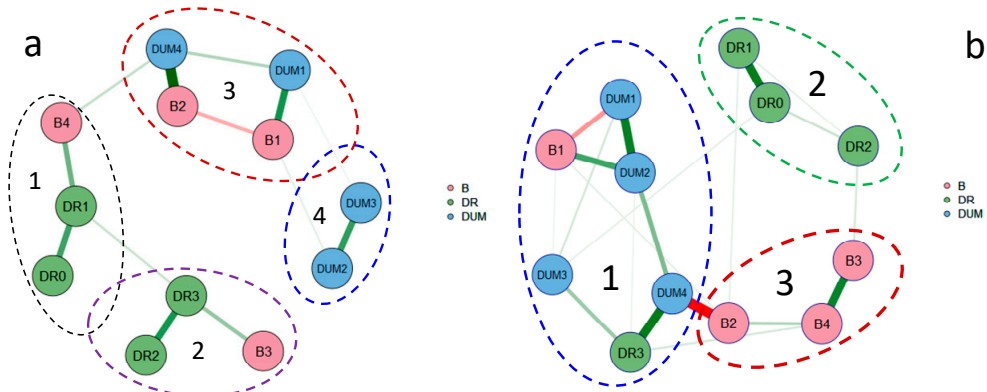

**Figure 10.** The network plots of the similarity (in R-statistics) between monitoring sites data of surface (**a**) and bottom (**b**) in the Ukrainian part of the Danube nearshore area in August 2018. The names of sampling stations as in Table 2. Sets of stations in the legend. The line thickness between stations reflects the correlation value; green is positive, red is negative. Clusters marked by dashed lines and numbered.

However, on the bottom horizon, we note a little bit different situation: B1 and DR3 sites were united with the dumping sites in one cluster (Figure 10b). This distribution is explained by the depth differences between DR2 and DR3, which is critical for phytoplankton in the bottom layer. The strong positive links on the bottom horizon were noted for the sites DR3 and DUM4; B3 and B4; DR1 and DR0.

Thus, we can see significant differences between the stations in their depth and type of anthropogenic impact. In general, the comparative statistical analysis showed differences in the strength and direction of the environmental impacts for the surface and the bottom. On the surface, it is possible to trace the flow from the impact of dredging on the background stations, whereas at the bottom, the background stations are well separated from the dredging and dumping sites, representing more stable conditions in these areas of the nearshore.

## 4. Discussion

The bioindication approach and ecological mapping help to reveal the main vectors of the interaction of environmental parameters and phytoplankton community in a very dynamic region of the Danube nearshore area. Application of these methods, based on nutrient concentrations and species bioindicators, was successfully tested for many water bodies and revealed the spatial dynamics and seasonal fluctuations of pollution in fresh and brackish waters [13,14,31,44]. For the analysis in the present study, we used 35 indicator species, which is 53% out of all found species in the study area of river-sea interaction. This testifies the necessity to find more indicator species in transition waters and the relevance of further research. It is extremely important to conduct a more detailed study of the species

diversity in the Danube nearshore area in order to identify the species that form the signal about certain changes.

The level of trophicity, which is determined by the concentration of nutrients and their transformation from one form to another, is the determining factor in the development of the phytoplankton assemblage in the studied region (Figure 9). The river-sea interaction results in the stratification of water masses: a surface layer with a salinity of 0.5–12‰ and a bottom layer formed by waters of marine genesis with a salinity of 16–18‰. This is an important aspect in such transition water areas located in the zone of river runoff influence. The formation of pelagic communities is determined by the complex hydrodynamic structure in the nearshore [2]. The main impact is the formation of a hydrofront—a zone of maximum horizontal gradients of water salinity. Its basic features are stable temperature stratification of water masses in summer and disturbance of diffuse exchange between the surface and bottom layers of the nearshore with the formation of bottom hypoxia. In addition, a great influence in the deltas is made by the rate of circulation of biogenic compounds of nitrogen and phosphorus with the participation of algal communities during production and destruction processes [45–47].

A peculiar feature of the hydrochemical regime of the Danube nearshore area is the imbalance of the ecosystem in terms of the content of the main biogenic elements—nitrogen and phosphorus, which is in line with previous research [2]. The balanced ecosystems are characterized with the ratio NT: PT = 7.2:1 (the standard ratio for an organic matter of plankton) [48]. The ratio of the mean values of NT and PT in the surface (44:1) and bottom (62:1) layers from the studied region also indicate the imbalance of the ecosystem with respect to nitrogen, which is associated with an excessive supply of mineral nitrogen compounds (nitrates) with river runoff (Figure 6).

Concentrations of phosphor compounds (DIP, DOP) and pH level testify to the higher trophic state in the region. It is known that DIP concentrations above 40 $\mu$g P L$^{-1}$ are typical for hypereutrophic waters [31]. The pH values are not usually considered to the trophic level, but in the cited classification system [31], it is given that high pH usually reflected the high trophicity of a waterbody. In the bottom water masses with higher salinity, there is no intensive accumulation of DOP and DON, which is associated with the active development of destruction processes and is confirmed by the lack of oxygen saturation of water in certain areas of the nearshore.

The non-Redfieldian stoichiometry is mentioned to be the character of many coastal and estuary water areas [49,50]. Under the eutrophic conditions, the biodiversity of the pelagic assemblages decreases; the input of diatoms lowers and is replaced by dinoflagellates and cyanobacteria [16]. The processes of enrichment with nutrients are accompanied by the development of small-cells green algae (in our case *Monoraphidium* sp.) and bigger cells of Miozoa (*Tripos* sp.). Such processes, associated with the long-term enrichment of the aquatic ecosystem with nutrients, lead to changes in biodiversity in the Danube River basin, the largest waterway in Europe [51].

In addition, the ratio of nitrogen and phosphorus is unbalanced, which was noted earlier in this region [2]. For example, in 2004–2010, the ratio of dissolved mineral and organic forms of nitrogen and phosphorus NT: PT in the photic layer in the nearshore area made 50:1; this is due to the input of nitrogen compounds from the dredging area being larger than that of phosphorus. Nitrogen compounds are mostly represented by their organic component—the share of DON in the total balance of nitrogen forms is more than 80% for the surface layer and about 90% for the bottom layer. The destruction of dissolved forms of organic nitrogen compounds, which leads to the formation of mineral forms (reduced, i.e., ammonia and oxidized, i.e., nitrites and nitrates), causes the further eutrophication of nearshore waters because the newly formed mineral nitrogen compounds enter the biotic turnover [1,2].

Calculated indices of saprobity and the state of the ecosystem, as well as the list of revealed species indicators with its ecological preferences, clearly demonstrate a dynamic and high trophicity of the studied area. According to the indicator species, the studied

waters are classified as medium eutrophic (beta-mesosaprobes), but the values of the index of saprobity show the water quality class 3 and 4 describing not only organic pollution but also the intensive self-purification processes. The WESI indices are significantly higher than 1 that also reflects the active work of the ecosystem both on the surface and in the bottom layers (Figure 6c,d). The same assessment of the state of the environment in the Danube Delta was obtained by other researchers [19]. However, in our case, a search and replenishment of species-indicators in the studied area should be the focus of future research.

As to the impact of the dumping and dredging effect, we may indicate that there is no clear understanding of this effect on the phytoplankton in the studied region, which is supported by other studies for zooplankton [52]. Some authors represent increasing in the autotroph's production [53], whereas another, it is diminishing [54]. Together with this, it was shown that the release of sediments during aggradation might cause the intensification of microalgae production [55]. The main factor influencing the algal community is the river inflow and the catchment area [56]. In this regard, the social-economic activities in the Danube River basin result in deltas and nearshore marine ecosystems; thus, sustainable planning should improve the water quality in the lower regions. The indicators of salinity (mesohalobes) had better describe the character of the transportation of fresh riverine waters than salinity gradient. This fact also supports the efficiency of the bioindication approach to reveal the patterns of plankton functioning in transition water bodies.

## 5. Conclusions

As a result of the analysis in August 2018, the 66 species of microalgae from six divisions were revealed in the Ukrainian part of the Danube nearshore area. More than half of the species were bioindicators with several ecological preferences: salinity, habitat, level of saprobity. Based on the index of saprobity, classes 3 and 4 of water quality were identified for the studied area. The biomass of phytoplankton was presented for 63.2% by diatoms (523.1 mg m$^{-3}$) and for 33.6% by Miozoa (303.2 mg m$^{-3}$). The NT:PT ratio varied from 34:1 to 47:1 and from 27:1 to 72:1 in the surface and bottom layers, respectively, which indicate the imbalance of the ecosystem with respect to nitrogen.

Thus, we confirmed our hypothesis about the effectiveness of bioindication by phytoplankton communities and statistical mapping to identify the impact of the Danube Delta waters on the Black Sea's coastal ecosystems, which was impossible to do only using chemical data. The use of statistical maps as an assessment model helped to identify the paths and range of penetration of the Danube waters into the coastal sea masses, and also showed the main trends in the transformation of river waters on the coast and ranked environmental factors in terms of their importance for the plankton community and the direction of impact. Two pools of factors were identified with oppositely directed influence on the development of the phytoplankton community: salinity and oxygen, on one hand, and nutrients, on the other. The nearshore area is a eutrophic ecosystem where the river is the main source of nutrients. Moreover, our results demonstrate that works on dredging and dumping additional cause entrance of the nitrogen, phosphorus, and siliceous. The imbalance of nitrogen and phosphorus and the velocity of its recycling were the leading factor of the processes of production and destruction. The most indicative groups of microalgae for the salinity and oxygen gradients were Miozoa and Chlorophyta.

The interaction of the Danube River mouth that has a great social-economic impact from most European countries, with the Black Sea, challenges the resilient approach in coastal and land-based management. Further application of the ecological mapping can help to assess the temporal and spatial dynamic of trophic relationships in the ecosystem in detail in transition waters, such as the Danube nearshore area.

**Author Contributions:** Conceptualization, A.S. and Y.B.; methodology, A.S., Y.B. and S.B.; software, S.B.; validation, A.S., Y.B. and S.B.; formal analysis, A.S., Y.B. and S.B.; investigation, A.S. and Y.B.; resources, non; data curation, A.S., Y.B. and S.B.; writing—original draft preparation, A.S., Y.B. and S.B.; writing—review and editing, A.S., Y.B. and S.B.; visualization, S.B.; supervision, Y.B. and S.B.; project administration, non; funding acquisition, non. All authors have read and agreed to the published version of the manuscript.

**Funding:** The work was done in the framework of scientific work of the Institute of marine biology of the NAS of Ukraine No 0119U000625 "Control monitoring observations during operation of the deep-water shipway the Danube–Black Sea (marine part)" and fundamental national scientific work "Patterns of the functioning of contour assemblages of the Black Sea ecosystems in view of disbalance of natural processes".

**Institutional Review Board Statement:** Not applicable.

**Informed Consent Statement:** Not applicable.

**Data Availability Statement:** Data sharing not applicable.

**Acknowledgments:** Authors are very grateful to the administration of the Institute of Marine Biology of the NAS of Ukraine for the organization of works in the Ukrainian part of the Danube Delta; to senior researcher of the Institute of Marine Biology of the NAS of Ukraine Eugeniy Sokolov for preparation of a map of the monitoring studies and to Maksim Martyniuk for his help in sampling. The authors also thank the Israeli Ministry of Aliyah and Integration.

**Conflicts of Interest:** The authors declare no conflict of interest.

## Appendix A

**Table A1.** Hydrochemical parameters in the Ukrainian part of Danube nearshore area in August 2018.

| Site | B1 | B2 | B3 | B4 | DR0 | DR1 | DR2 | DR3 | DUM1 | DUM2 | DUM3 | DUM4 |
|---|---|---|---|---|---|---|---|---|---|---|---|---|
| Surface | 1 | 2 | 3 | 4 | 5 | 6 | 7 | 8 | 9 | 10 | 11 | 12 |
| North | 45.35 | 45.27 | 45.27 | 45.29 | 45.34 | 45.34 | 45.32 | 45.32 | 45.32 | 45.32 | 45.30 | 45.32 |
| East | 29.87 | 29.86 | 29.82 | 29.80 | 29.76 | 29.77 | 29.79 | 29.81 | 29.86 | 29.88 | 29.83 | 29.86 |
| Depth, m | 21.0 | 25.0 | 19.3 | 14.0 | 6.6 | 5.5 | 4.0 | 9.8 | 19.0 | 23.0 | 19.8 | 17.2 |
| pH | 9.09 | 8.90 | 8.87 | 8.87 | 8.39 | 8.50 | 8.45 | 8.86 | 9.10 | 9.00 | 9.12 | 8.84 |
| Secchi, m | 5.0 | 7.5 | 1.0 | 0.9 | 0.5 | 0.5 | 0.5 | 1.0 | 4.5 | 4.0 | 3.0 | 5.5 |
| TSS, mg $L^{-1}$ | 9.10 | 8.78 | 7.59 | 43.20 | 130.00 | 120.00 | 111.00 | 50.20 | 9.15 | 7.25 | 10.70 | 7.70 |
| Salinity,‰ | 14.88 | 16.34 | 10.53 | 8.16 | 0.17 | 0.23 | 0.34 | 7.51 | 14.55 | 16.74 | 12.22 | 16.54 |
| Oxygen, mg $L^{-1}$ | 9.80 | 7.99 | 7.96 | 7.17 | 8.32 | 7.86 | 8.59 | 7.54 | 11.00 | 9.92 | 11.20 | 9.06 |
| $NH_4^+$, µg N $L^{-1}$ | 0 | 0 | 0 | 53.5 | 0 | 0 | 17.6 | 0 | 25.3 | 0 | 0 | 0 |
| $NO_2^-$, µg N $L^{-1}$ | 2.11 | 0.70 | 8.39 | 9.43 | 24.37 | 13.87 | 16.23 | 12.31 | 1.59 | 1.59 | 3.14 | 0.92 |
| $NO_3^-$, µg N $L^{-1}$ | 16.90 | 79.50 | 31.80 | 417.00 | 1490.00 | 1390.00 | 765.00 | 477.00 | 39.20 | 22.80 | 112.00 | 1.99 |
| $PO_4^{3-}$, µgP $L^{-1}$ | 5.12 | 3.92 | 10.90 | 25.70 | 70.10 | 66.20 | 59.20 | 24.80 | 4.52 | 7.54 | 7.24 | 5.73 |
| $SiO_3$, µg $L^{-1}$ | 93 | 13 | 794 | 1060 | 2834 | 2936 | 2724 | 1189 | 103 | 93 | 252 | 64 |
| **Bottom** | **B1** | **B2** | **B3** | **B4** | **DR0** | **DR1** | **DR2** | **DR3** | **DUM1** | **DUM2** | **DUM3** | **DUM4** |
| pH | 8.30 | 8.33 | 8.88 | 8.81 | 8.39 | 8.44 | 8.72 | 8.83 | 8.73 | 8.32 | 8.34 | 8.70 |
| Secchi, m | 5.0 | 7.5 | 1.0 | 0.9 | 0.5 | 0.5 | 0.5 | 1.0 | 4.5 | 4.0 | 3.0 | 5.5 |
| TSS, mg $L^{-1}$ | 11.80 | 10.80 | 4.16 | 53.70 | 130.00 | 116.00 | 170.00 | 14.60 | 9.33 | 16.50 | 10.30 | 8.07 |
| Salinity, ‰ | 18.02 | 18.06 | 18.06 | 17.22 | 0.23 | 0.27 | 11.21 | 16.50 | 18.13 | 18.50 | 17.90 | 17.20 |
| Oxygen, mg $L^{-1}$ | 5.63 | 3.52 | 7.82 | 7.16 | 8.32 | 8.43 | 8.41 | 7.52 | 10.60 | 5.47 | 5.01 | 7.96 |
| $NH_4^+$, µg N $L^{-1}$ | 15.0 | 21.1 | 15.0 | 15.0 | 15.0 | 15.0 | 15.0 | 15.0 | 15.0 | 15.0 | 15.0 | 15.0 |
| $NO_2^-$, µg N $L^{-1}$ | 5.80 | 4.03 | 1.07 | 2.40 | 10.54 | 16.97 | 26.29 | 2.11 | 1.00 | 4.18 | 5.36 | 0.63 |
| $NO_3^-$, µg N $L^{-1}$ | 33.80 | 19.90 | 9.93 | 39.70 | 1371.00 | 1439.00 | 1281.00 | 69.50 | 1.49 | 27.80 | 27.80 | 0.99 |
| $PO_4^{3-}$, µgP $L^{-1}$ | 10.60 | 12.40 | 3.92 | 12.40 | 63.20 | 80.40 | 46.20 | 5.73 | 5.12 | 7.24 | 8.75 | 5.73 |
| $SiO_3$, µg $L^{-1}$ | 964 | 741 | 123 | 323 | 2885 | 2936 | 1401 | 745 | 95 | 822 | 143 | 95 |

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
