# Peer review of "Assessment of River-Sea Interaction in the Danube Nearshore Area (Ukraine) by Bioindicators and Statistical Mapping"

_land, doi:10.3390/land10030310_

Round 1

Reviewer 1 Report

My major comments are as follows:

  1. In my sense, “ecological mapping” refers to a map that describes the spatial distribution of a certain ecological state, whereas this manuscript shows simply spatial patterns in several variables. I don’t feel “ecological mapping” or some other wards related to this are appropriate for this manuscript.

  1. Introduction: the authors separated paragraphs even if these paragraphs describe the same topic - some paragraphs should be merged into one paragraph. Also, connections between the paragraphs are unclear. These issues make this section difficult to follow for many readers.

  1. Materials ad Methods: the authors should state more deeply about their approaches used in this study, particularly about the water quality classification technique. I understand the readers can understand if the readers look into the cited papers, but stating the approaches in this manuscript would be helpful.

  1. Results: I don’t understand what the lines 303–310 say. How did the authors distinguish the cluster? Or it just refers to the group? I guess the authors wanted to say there are two groups; one reacted to salinity and transparency, another reacted to nutrients and oxygen. If so, how do the authors explain about TSS, No Spec, and WESI – these variables seem to be more important in determining phytoplankton community structures than oxygen.

Discussion: The discussion section extends to general topics that are mostly extraneous to the results presented in this manuscript. The discussion section instead should focus on why and how water quality and phytoplankton community varied in the study area. The authors should describe carefully the connections between the results they got – the connections remain unclear in this manuscript. Also, the authors didn’t discuss how river-sea interaction influences water quality and phytoplankton community, although this manuscript is entitled as “Assessment of river-sea interaction in the Danube nearshore area by ecological mapping.” I encourage the authors to rewrite the discussion section to array what they got in the present study.

Author Response

The authors are very thankful to all the reviewers for their time and attention to our work. We tried to take into consideration all the comments and remarks.

Answer to the reviewer 1:

1.     In my sense, “ecological mapping” refers to a map that describes the spatial distribution of a certain ecological state, whereas this manuscript shows simply spatial patterns in several variables. I don’t feel “ecological mapping” or some other wards related to this are appropriate for this manuscript.

The title is changed to:

Assessment of river-sea interaction in the Danube nearshore area (Ukraine) by bioindicators and statistical mapping

2.     Introduction: the authors separated paragraphs even if these paragraphs describe the same topic - some paragraphs should be merged into one paragraph. Also, connections between the paragraphs are unclear. These issues make this section difficult to follow for many readers. 

Actually, we do not see which paragraphs may be united. They will be too long and reading will be more difficult.

Each paragraph one by one describes the situation in the studied region: general remarks about the hydrochemical peculiarities, the salinity regime; than the condition in the navigable arm; later the previous studies on phytoplankton; bioindicator and statistical approach that we would like to apply in our research; and finally, or aim and tasks.

Sorry, but we can`t see what to change in this section

3.     Materials ad Methods: the authors should state more deeply about their approaches used in this study, particularly about the water quality classification technique. I understand the readers can understand if the readers look into the cited papers, but stating the approaches in this manuscript would be helpful.

We added the table (Table 1 – line 143) that describe the scale of water quality based on Index of saprobity.

4.     Results: I don’t understand what the lines 303–310 say. How did the authors distinguish the cluster? Or it just refers to the group? I guess the authors wanted to say there are two groups; one reacted to salinity and transparency, another reacted to nutrients and oxygen. If so, how do the authors explain about TSS, No Spec, and WESI – these variables seem to be more important in determining phytoplankton community structures than oxygen.

We added some explanation in the text (340-343 lines). In addition, TSS is united with the nutrients both on bottom and surface layers (344-345 lines). No of Spec is discussed separately, because this is not the environmental parameter as it is (334-335 lines).

Discussion: The discussion section extends to general topics that are mostly extraneous to the results presented in this manuscript. The discussion section instead should focus on why and how water quality and phytoplankton community varied in the study area. The authors should describe carefully the connections between the results they got – the connections remain unclear in this manuscript. Also, the authors didn’t discuss how river-sea interaction influences water quality and phytoplankton community, although this manuscript is entitled as “Assessment of river-sea interaction in the Danube nearshore area by ecological mapping.” I encourage the authors to rewrite the discussion section to array what they got in the present study.

Some changes were done in discussion (397-398; 404-405; 407; 418 lines).

We noted (104-106 lines) that this is the first attempt to assess the river-sea interaction by mentioned methods. To have more clear understanding the interannual comparison will be made in future and will be based on present research.

Reviewer 2 Report

This is a well-written paper and quite informative. I think the results described here have significant local value. The paper detects patterns in several hydro-bio-chemical parameters in a coastal zone of Ukraine. The results are well-reported and thoroughly explained. If possible (not mandatory, a suggestion), perhaps consider plotting the statistically generated maps as GIS layers instead of plotting them over East and North coordinates. Also, by combining several statistical layers, it might be possible to generate certain hotspots for the region – such an analysis would be interesting to look at. Spatial correlation maps between the layers might do the trick.

For the introduction, it would be good to start by providing a paragraph that explains the problem statements on a regional scale (e.g., European countries). Perhaps explain the problem first and then narrow it down to the extent of the problem for the case study.

There are few typos or grammar issues that need to be fixed, however, nothing major.

Author Response

The authors are very thankful to all the reviewers for their time and attention to our work. We tried to take into consideration all the comments and remarks.

This is a well-written paper and quite informative. I think the results described here have significant local value. The paper detects patterns in several hydro-bio-chemical parameters in a coastal zone of Ukraine. The results are well-reported and thoroughly explained. If possible (not mandatory, a suggestion), perhaps consider plotting the statistically generated maps as GIS layers instead of plotting them over East and North coordinates. Also, by combining several statistical layers, it might be possible to generate certain hotspots for the region – such an analysis would be interesting to look at. Spatial correlation maps between the layers might do the trick.

Thank you, we will consider this in future

For the introduction, it would be good to start by providing a paragraph that explains the problem statements on a regional scale (e.g., European countries). Perhaps explain the problem first and then narrow it down to the extent of the problem for the case study.

We added the first paragraph (lines 41-46)

There are few typos or grammar issues that need to be fixed, however, nothing major.

Checked and done some changes

Reviewer 3 Report

Only a few comments for this interesting manuscript:

Line 28- 29: To arrange the keywords in the alphabetic order.

Lines 71 - 72: I think that you should add some recent references to support your sentence “using different approaches to assessing the quality of the aquatic environment.”. I would like to suggest:

Bosso, L., De Conno, C., & Russo, D. (2017). Modelling the risk posed by the zebra mussel Dreissena polymorpha: Italy as a case study. Environmental Management60(2), 304-313.

Jha, M. K., Shekhar, A., & Jenifer, M. A. (2020). Assessing groundwater quality for drinking water supply using hybrid fuzzy-GIS-based water quality index. Water research179, 115867.

Lines 100 - 139: Please, explain better your methods. This section is useless if written in this way. You absolutely must provide all the information on how you did the experiments!

Results and discussion: Well written!

Author Response

The authors are very thankful to all the reviewers for their time and attention to our work. We tried to take into consideration all the comments and remarks.

Only a few comments for this interesting manuscript:

Line 28- 29: To arrange the keywords in the alphabetic order.

Response: The key words are arranged in the alphabetic order

Lines 71 - 72: I think that you should add some recent references to support your sentence “using different approaches to assessing the quality of the aquatic environment.”. I would like to suggest:

Bosso, L., De Conno, C., & Russo, D. (2017). Modelling the risk posed by the zebra mussel Dreissena polymorpha: Italy as a case study. Environmental Management60(2), 304-313.

Jha, M. K., Shekhar, A., & Jenifer, M. A. (2020). Assessing groundwater quality for drinking water supply using hybrid fuzzy-GIS-based water quality index. Water research179, 115867.

Response: The references added – lines 75-76

Lines 100 - 139: Please, explain better your methods. This section is useless if written in this way. You absolutely must provide all the information on how you did the experiments!

Response: The description of the methods of eater quality assessment are added to this section

Results and discussion: Well written!

Reviewer 4 Report

Dear Authors,

the paper deals with an interesting and current topic, which represents one of the innovations by ecological mapping.

The paper is well written, has a clear structure and the purpose is well defined.

Only issues with the paper is that there is missing the clear relation between hypothesis and methodology used. No hypotheses, no research questions are defined at all. The discussion is more in the form of a conclusion.

Author Response

The authors are very thankful to all the reviewers for their time and attention to our work. We tried to take into consideration all the comments and remarks.

Dear Authors,

the paper deals with an interesting and current topic, which represents one of the innovations by ecological mapping.

The paper is well written, has a clear structure and the purpose is well defined.

Only issues with the paper is that there is missing the clear relation between hypothesis and methodology used. No hypotheses, no research questions are defined at all. The discussion is more in the form of a conclusion.

We added hypothesis and its clarification in the discussion: 110-112 lines; 484-491 lines

Round 2

Reviewer 1 Report

This paper is an important contribution and I recommend that it be accepted for publication.